# Bioactive Phytochemicals and Quenching Activity of Radicals in Selected Drought-Resistant *Amaranthus tricolor* Vegetable Amaranth

**DOI:** 10.3390/antiox11030578

**Published:** 2022-03-17

**Authors:** Umakanta Sarker, Shinya Oba, Sezai Ercisli, Amine Assouguem, Amal Alotaibi, Riaz Ullah

**Affiliations:** 1Department of Genetics and Plant Breeding, Faculty of Agriculture, Bangabandhu Sheikh Mujibur Rahman Agricultural University, Gazipur 1706, Bangladesh; 2Laboratory of Field Science, Faculty of Applied Biological Sciences, Gifu University, Yanagido 1-1, Gifu 501-1193, Japan; soba@gifu-u.ac.jp; 3Department of Horticulture, Agricultural Faculty, Ataturk University, TR-25240 Erzurum, Turkey; sercisli@gmail.com; 4Laboratory of Functional Ecology and Environment, Faculty of Sciences and Technology, Sidi Mohamed Ben Abdellah University, Imouzzer Street, Fez 2202, Morocco; assougam@gmail.com; 5Department of Basic Science, College of Medicine, Princess Nourah Bint Abdulrahman University, P.O. Box 84428, Riyadh 11671, Saudi Arabia; amaalotaibi@pnu.edu.sa; 6Department of Pharmacognosy, College of Pharmacy, King Saud University, Riyadh 11451, Saudi Arabia; rullah@ksu.edu.sa

**Keywords:** drought resistant, *A. tricolor*, proximate composition, nutraceuticals, phytopigments, vitamins, polyphenols, flavonoids, antioxidant activity, DPPH, ABTS^+^

## Abstract

Leafy vegetables are susceptible to drought stress. *Amaranthus tricolor* vegetables are resistant to abiotic stress, including drought, and are a source of ample natural phytochemicals of interest to the food industry due to their benefits to consumers’ health. Hence, the selected drought-resistant amaranth genotypes were evaluated for phytochemicals and antioxidant activity in an RCBD study with three replicates. The selected drought-resistant amaranth accessions contained ample carbohydrates, protein, moisture, and dietary fiber. We noticed many macroelements and microelements including iron, copper, manganese, zinc, sodium, molybdenum, boron, potassium, calcium, magnesium, phosphorus, and sulfur; adequate phytopigments, including betacyanins, betalains, betaxanthins, carotenoids, and chlorophylls; plentiful bioactive phytochemicals, including ascorbic acid, flavonoids, polyphenols, and beta-carotene; and antioxidant potential in the selected drought-resistant amaranth accessions. The drought-resistant amaranth accessions VA14 and VA16 were proven to have high ascorbic acid, beta-carotene, and polyphenol levels. The drought-resistant accessions VA12 and VA14 had high flavonoid levels. The drought-resistant accessions VA3, VA14, and VA16 had high AC both in regard to both DPPH and ABTS^+^. These drought-resistant accessions, VA3, VA14, and VA16, can be utilized as high-yielding varieties with antioxidant profiles for purposes of drinks. The correlation study revealed that bioactive phytopigments and phytochemicals of amaranth accessions had good free radical quenching capacity against 2,2′-azino-bis (3-ethylbenzothiazo-6-sulfonic acid) and diphenyl-1-picrylhydrazyl, equivalent to Trolox. It was revealed in the present study that these drought-resistant accessions contain plentiful proximate, nutraceuticals, phytopigments, bioactive phytochemicals, and antioxidant potentiality. Their drought resistance and quenching of ROS offer huge prospects for the promotion of health benefits and the feeding of communities in drought-prone semiarid and arid areas of the globe, especially those deficient in nutraceuticals, phytopigments, and antioxidants.

## 1. Introduction

The color, flavor, and taste of the foods predominantly influence the acceptance of food by consumers. Currently, tasty and colored food products are popular around the globe. Furthermore, food can attract consumers due to its nutritional, safety, and beauteousness. There is a great surge in demand for natural phytopigments, including betacyanins, anthocyanin, betaxanthins, amaranthine, betalains, carotenoids, and chlorophylls. *A. tricolor* is the sole source of betalains, betaxanthins, and betacyanins among leafy vegetables that have high quenching activity against free radicals [1,2,3]. Betalains can be widely utilized as colorants and additives for foods, especially in low-acid foods. Its stability at different pHs is better than that of anthocyanins [4]. The critical pigment among the betacyanins is amaranthine, which is a strong antioxidant. Amaranthine can be utilized as an additional source of betanins from red beets, as these can be used in food coloring or as an antioxidative additive [1]. Amaranth is extensively acclimated to abiotic stresses, including drought [5,6,7,8] and salinity [9,10,11,12].

*A. tricolor* is a rapid-growing plant with a C_4_ pathway and several uses, including as a vegetable, as a grain, and in decoration. This extensively acclimated crop is distributed throughout the globe, including Asia, Australia, Africa, America, and Europe. *A. tricolor* is one of the cheapest leafy vegetables and has copious protein containing essential amino acids, including methionine and lysine; vitamin C; dietary fiber; carotenoids; and minerals [13,14,15,16,17,18]. Leafy vegetables of this species have ample phytopigments, including betalains, anthocyanins, betacyanins, carotenoids, betaxanthins, and chlorophylls which have a high capacity for quenching radicals [19,20]. They also have other bioactive phytochemicals, such as vitamin C, phenols, and flavonoids [21]. Due to their antioxidative effect in the human body, these bioactive compounds are substantially involved in food manufacturing through defending numerous illnesses, including cardiovascular disease, cataracts, emphysema, cancer, atherosclerosis, retinopathy, arthritis, and neurodegeneration [22].

Among red and green amaranth, red amaranth contains more phytopigments, including betacyanins, amaranthine, betaxanthins, anthocyanin, carotenoids, and betalains. The leafy amaranth has pronounced plasticity and variability in Bangladesh, and in Asia and elsewhere. The selected bright-red drought-resistant *A. tricolor* vegetable accessions have plentiful betalains. The gorgeous color, flavor, and taste make it a standard leafy vegetable in the Asian continent, including many regions of the globe. It can be grown year-round in Bangladesh, including the off-period for leafy vegetables from winter to summer [13,14]. Leaves of amaranth prohibited the proliferation of colorectal (Caco-2), liver (HepG2), and breast (MCF-7) cancer cells, and exhibited antitumor potential [23].

We studied bioactive phytopigments of leafy vegetables, including bioactive phytochemicals, as these bioactive compounds have great involvement in the food industry. In our earlier studies, amaranth germplasms were screened for high yields, antioxidant potentiality, and drought resistance based on morphological and physiological parameters. From these studies, we selected the best four drought-resistant amaranth accessions (VA3, VA12, VA14, and VA16). We evaluated the bioactive phytopigments, nutraceuticals, antioxidant capacities, and bioactive phytochemicals in the selected drought-resistant *A. tricolor* accessions. We have been exploring drought-resistant amaranth genotypes as sources of natural phytopigments because of their ample betacyanins, betaxanthins, and betalains, which are nutraceuticals and phytochemicals of interest in the food industry for their benefits to consumers’ health, especially in drought-affected semiarid and arid areas of the world. We hypothesized that the selected high-yielding drought-resistant accessions have abundant phytochemicals with high radical scavenging capacity. Therefore, we examined the nutraceuticals, bioactive phytochemicals, phytopigments, and antioxidant potentiality of each drought-resistant amaranth.

## 2. Materials and Methods

### 2.1. Experimental Materials

We select the best four antioxidant potential high-yielding accessions from 43 accessions evaluated previously.

### 2.2. Design and Layout

Completely randomized block design (RCBD) in three replicates was followed to execute the study at Bangabandhu Sheikh Mujibur Rahman Agricultural University. A 20 cm and 5 cm between rows and the plant’s distance in each genotype was followed to grown accessions in a 1 m^2^ experimental plot.

### 2.3. Intercultural Practices

We applied recommended fertilizer, compost doses and maintained appropriate cultural practices [24]. The row spacing was maintained precisely by thinning the plants grown in the row. Weeding and hoeing were done at regular intervals to remove the weeds of experimental plots properly. Regular irrigation was provided to ensure normal growth. Leaf samples were collected 30 days after the sowing of plants.

### 2.4. Solvent and Reagents

Solvent: Acetone, methanol, and hexane. Reagents: HClO_4_, dithiothreitol (DTT), cesium chloride, ascorbic acid, H_2_SO_4_, HNO_3_, Folin–Ciocalteu reagent, Trolox (6-hydroxy-2, 5,7,8-tetramethyl-chroman-2-carboxylic acid), aluminum chloride hexahydrate, rutin, gallic acid, potassium acetate, DPPH, ABTS^+^, 2,2-dipyridyl, sodium carbonate, and potassium persulfate.

### 2.5. Estimation of Proximate Composition

Moisture, crude protein contents, fiber, crude fat, gross energy, and ash were estimated from leaves following AOAC method [25]. The micro-Kjeldahl method was followed to calculate the nitrogen. Finally, crude protein was measured by the multiplication of nitrogen with 6.25. We subtracted the moisture, crude fat, crude protein, and ash (%) from 100 to calculate carbohydrate (g 100 g^−1^ FW).

### 2.6. Estimation of Mineral Composition

An oven was used to dry the leaves for 24 h at 70 °C. Then, leaves were ground in a mill. We digested the sample in nitric acid and perchloric acid to determine calcium, magnesium, potassium, iron, copper, zinc, and manganese [25]. Briefly, in the presence of carborundum beads, 10 mL H_2_SO_4_ (96%), 40 mL HclO_4_ (70%), and 400 mL HNO_3_ (65%) were added to the 0.5 g samples. Then, p was determined in triplicate through dilution of the solution appropriately following the ascorbic acid method. The ascorbic acid and antimony were added to the yellow-colored complex solution for converting it to a blue-colored phosphomolybdenum complex. The absorbance was taken by atomic absorption spectrophotometry (AAS) (Hitachi, Tokyo, Japan) at wavelength of 285.2 nm (magnesium), 76 6.5 nm (potassium), 880 nm (phosphorus), 248.3 nm (iron), 258.056 nm (sulfur), 279.5 nm (manganese), 213.9 nm (zinc), 422.7 nm (calcium), 589 nm (sodium), 313.3 nm (molybdenum), 430 nm (boron), and 324.8 nm (copper).

### 2.7. Determination of Chlorophylls and Carotenoids

Chlorophyll *ab*, chlorophyll *b*, carotenoids, and chlorophyll *a* were calculated by extracting the leaves in acetone (80%) [26]. The absorbance was detected using a spectrophotometer (Hitachi, Tokyo, Japan) at 646, 663, and 470 nm. Finally, chlorophylls were calculated as micrograms per gram and total carotenoid milligrams per 100 g of fresh weight.

### 2.8. Measurement of Betaxanthins and Betacyanins Content

The method of Sarker and Oba [27] was followed to measure betaxanthins and betacyanins. We extracted the leaf samples in 80% MeOH containing 50 mM ascorbate. A spectrophotometer (Hitachi, Tokyo, Japan) was used to read the absorbance at 475 and 540 nm for betaxanthins and betacyanins, respectively. We calculated the data as ng indicaxanthin and betanin g^−1^ of fresh weight (FW) for betaxanthins and betacyanins.

### 2.9. Beta-Carotene Estimation

Our previously described method was followed to estimate beta-carotene [28]. In 10 mL acetone (80%), a 500 mg leaf sample (fresh) was added by grinding meticulously in a mortar and pestle. We centrifuged the extract at 10,000× *g* for 3–4 min. After removing the supernatant, the final volume was marked up to 20 mL in a volumetric flask. The absorbance was taken using a spectrophotometer (Hitachi, Tokyo, Japan) at 480 and 510 nm, respectively. The results were calculated as mg 100 g^−1^ FW.
Beta-carotene = 7.6 (Abs. at 480) − 1.49 (Abs. at 510) × Final volume/(1000 × fresh weight of leaf)

### 2.10. Estimation of Ascorbic Acid

Dehydroascorbic acid (DHA) and ascorbic acid (AsA) were determined from the leaves (fresh) using a spectrophotometer (Hitachi, Tokyo, Japan). Pre-incubation of sample and reduction of dehydroascorbic acid into ascorbic acid was performed using Dithiothreitol (DTT). Ascorbic acid reduced F^3+^ to F^2+^. Reduced F^2+^ and 2, 2-dipyridyl produced complexes [28]. A spectrophotometric (Hitachi, Japan) was used to estimate ascorbic acid by reading the optical density of Fe^2+^ complexes at 525 nm. Ascorbic acid was calculated in mg per 100 g of FW.

### 2.11. Samples Extraction for TPC, TFC, and TAC Analysis

For total polyphenol content, total flavonoid content, and total antioxidant activity determination, the fresh leaves were harvested to make the extraction. The leaves were harvested 30 days after sowing and dried overnight. The samples were grounded with mortar and pestle for chemical analysis. 0.25 g of leaf powder was dissolved in 10 mL 90% methanol in a bottle capped tightly. The bottle was set in a water bath (Thomastant T-N22S, Thomas Kagaku Co. Ltd., Tokyo, Japan) with shaking. After 1 h, the extract was filtered for further analytical assays of polyphenol content, flavonoid content, and antioxidant activity.

### 2.12. Determination of Total Polyphenols Content

Phenolic content was determined by the Folin–Ciocalteu reagent [28]. In a test tube, 50 μL of the fresh leaf extract solution was placed. Then, 1 mL of Folin–Ciocalteu reagent (previously diluted with distilled water; reagent: water = 1:4) was added, and the contents were mixed meticulously. One milliliter of Na_2_CO_3_ (10%) was added after 3 min, and the mixture was allowed to stand for one hour in the dark. The optical density was taken at 760 nm using a spectrophotometer (Hitachi, Tokyo, Japan). The concentration of total phenolic compounds in leaf extracts was determined as μg g^−1^ of gallic acid equivalents using an equation (Y = 0.009X + 0.019) obtained from a standard gallic acid graph and the following formula:C = (c × v)/m,
where C = total phenolic content, c = concentration of gallic acid, v = volume of extract and m = weight of crude extract in g. Results are expressed in equivalents of gallic acid (GAE) standard µg g^−1^ of FW.

### 2.13. Determination of Total Flavonoid Content

The aluminum chloride colorimetric method was followed to estimate the total flavonoid content [28]. For this, 500 μL of leaf extract was placed in a test tube. Then, 0.1 mL of 1M potassium acetate, 2.8 mL of distilled water, 0.1 mL of 10% aluminum chloride, and 1.5 mL of methanol were added. The absorbance was measured at 415 nm using a spectrophotometer (Hitachi, Tokyo, Japan) after allowing the mixture to stand for 30 min at room temperature. Rutin was used as a standard compound to make the standard graph (Y = 0.013X) using the following formula:C = (c × v)/m,
where C = total flavonoid content, c = concentration of rutin, v = volume of extract and m = weight of crude extract in g. Results are expressed as μg g^−1^ rutin equivalents of dry weight (dw).

### 2.14. Radical Quenching Capacity Assay

The diphenyl-picrylhydrazyl (DPPH) radical degradation method was followed to estimate the antioxidant activity [29]. A 1 mL portion of DPPH solution (250 µM) and 4 mL of distilled water were added to leaf extract (10 µL) in a test tube (in triplicate). The mixture was kept in a dark place for 30 min. A Hitachi spectrophotometer (Tokyo, Japan) was used to take the absorbance at 517 nm. ABTS^+^ solutions (7.4 mM) and potassium persulfate (2.6 mM) were used to make the stock solutions for ABTS^+^ assay [30]. Two stock solutions were equally mixed to prepare the working solution. In a dark place, the mixed solution was stood for 12 h to react at room temperature. Leaf extract (150 μL) was poured to ABTS^+^ solution (2850 μL) (1:60 *v*/*v* ABTS^+^ solution: MeOH) and allowed to stand for 2 h in the dark. A Hitachi spectrophotometer (Tokyo, Japan) was used to read the optical density against MeOH at 734 nm. Percentage inhibition of DPPH and ABTS^+^ against control was utilized to determine the antioxidant activity following the equation:% Antioxidant activity = (Ac − A_ls_/Ac) × 100
where Ac is the optical density of the control (10 µL and 150 μL MeOH for TAC (DPPH) and TAC (ABTS) in its place of leaf extract) and A_ls_ is the absorbance of leaf samples. Trolox is the standard compound, and the results were converted to μg TE g^−1^ DW.

### 2.15. Statistical Analysis

The sample data were averaged replication-wise to obtain the replication means. One-way analysis of variance (ANOVA) was calculated using Statistix 8 software [31,32]. The mean comparison was performed by Duncan’s Multiple Range Test (DMRT) at a 1% level of probability. The results are displayed as mean ± SD.

## 3. Results and Discussion

Analysis of variance (ANOVA) provided significant variations among treatments for all the considered traits. A wide range of variations were also reported in the agronomic traits of maize [33,34,35], rice [36,37,38,39,40,41,42,43,44,45,46,47,48,49,50], okra [51,52,53], broccoli [54], and coconut [55,56].

### 3.1. Composition of Proximate

Figure 1 represents the proximate in the four drought-resistant *A. tricolor* accessions. The moisture content significantly varied from 81.52 g 100 g^−1^ in VA16 to 88.16 g 100 g^−1^ VA3 on a fresh weight basis. As lower moisture content ensured higher dry matter of the leaves, two drought-resistant accessions, VA14 and VA16, had high dry biomass (approximately 18–19% dry matter). Leaf moisture was linearly correlated with maturity. The results are corroborative of the previous reports of Sarker and Oba [5] and of the results in sweet potato leaves found by Sun et al. [57], respectively. The content of protein had pronounced and significant variations in the leaves of four drought-resistant *A. tricolor* accessions, from 2.15 g 100 g^−1^ in the drought-resistant accession VA3 to 6.21 g 100 g^−1^ in the drought-resistant accession VA14. The protein requirements of poor people and vegetarians in developing countries are mainly fulfilled from leafy vegetables, including amaranth. The amounts of protein obtained from this drought-resistant amaranth were much higher than we found in a previous study of *A. tricolor* (1.26%) [14]. This difference might have been due to the differences in the genetic makeup of accessions, cultural practices, and the edaphic and climatic conditions of the growing season. The fat content had significant variations in the selected drought-resistant amaranths. The content of fat varied from 0.35 to 0.43 g 100 g^−1^ fresh weight (FW). That finding can be corroborated by our previous study [5] and the results about the leaves of sweet potato by Sun et al. [57]. They noticed that fat’s impacts include the organs, cell function, and maintenance of temperature in the body. Fats included ample fatty acids, including omega-3 and omega-6. Fats contribute to the absorption and transport of vitamins soluble in fats (D, E, K, and A) and digestion.

All parameters revealed prominent and significant variations among drought-resistant amaranths. The drought-resistant accession VA16 had the maximum contents of carbohydrates (8.38 g 100 g^−1^ FW), digestible fiber (7.82 g 100 g^−1^ FW), and energy (56.61 kcal 100 g^−1^ FW). Thereafter came accessions VA12, and VA14. The carbohydrates (5.73 g 100 g^−1^ FW) and energy (33.60 kcal 100 g^−1^ FW) were least in VA3, and the dietary fiber content (2.68 g 100 g^−1^ FW) was the lowest in VA12. The accession VA12 had the maximum ash content (7.22 g 100 g^−1^ FW), followed by VA3. VA16 displayed the lowest ash content (4.45 g 100 g^−1^ FW), beaten by VA14. Dietary fiber is significantly involved in palatability and digestibility—preventing constipation [16]. It was observed from this investigation that drought-resistant amaranth leaves have considerable amounts of digestible fiber moisture, protein, and carbohydrates. The findings corroborate our previous findings [5]. The carbohydrate contents observed in the drought-resistant accession VA16 were higher than the equivalent results in our previous studies of red morph amaranth [58], *A. spinosus* (weedy amaranth) [59], green morph amaranth [60], stem amaranth [61], and *A. blitum* [62]. The digestible fiber obtained from the drought-resistant accession VA16 can be corroborated by our previous studies of red morph amaranth [58], green morph amaranth [60], stem amaranth [61], and *A. blitum* [62]. However, the dry matter contents of these drought-resistant accessions VA14 and VA16 were greater than the dry matter contents of red morph amaranth [58], weedy amaranth [59], green morph amaranth [60], stem amaranth [61], and *A. blitum* [62]. The protein contents of these drought-resistant accessions VA14 and VA16 were greater than the protein contents of red morph amaranth [58], green morph amaranth [60], stem amaranth [61], and *A. blitum* [62].

### 3.2. Macro and Microelements

The macro and microelements in the selected drought-resistant amaranths are shown in Figure 2 and Figure 3, respectively. The drought-resistant accessions showed remarkable variations in calcium (1.72 to 3.28 mg g^−1^ FW), phosphorus (0.62 to 1.81 mg g^−1^ FW), and sulfur (0.55 to 1.34 mg g^−1^ FW) content. The drought-resistant accession VA14 had the highest contents of potassium (7.56 mg g^−1^ FW), calcium (3.28 mg g^−1^ FW), phosphorus (1.81 mg g^−1^ FW), and sulfur (1.34 mg g^−1^ FW), albeit the maximum magnesium content (3.27 mg g^−1^ FW) was noted in VA16. On the other hand, VA12 had the lowest potassium (4.33 mg g^−1^ FW), phosphorus (0.62 mg g^−1^ FW), sulfur (0.55 mg g^−1^ FW), and magnesium (2.50 mg g^−1^ FW) contents. Accession VA16 had the lowest calcium content (1.72 mg g^−1^ FW) (Figure 2). We obtained ample potassium, calcium, magnesium, phosphorus, and sulfur in the drought-resistant amaranth. Chakrabarty et al. [18] in *A. lividus* also demonstrated the same results. In the literature, several amaranth species [63] reported ample magnesium, potassium, phosphorus, calcium, and sulfur. They also mentioned that amaranth’s magnesium, potassium, phosphorus, calcium, and sulfur contents were much more pronounced than spider flower, kale, spinach, and black nightshade.

The drought-resistant amaranth had ample microelements with prominent variations among accessions (Figure 3). The iron, copper, zinc, boron, manganese, molybdenum, and sodium varied as follows: 10.26–17.41, 0.98–2.29, 10.58–14.63, 5.18–7.44, 10.23–16.81, 0.26–0.58, and 62.55–80.32 µg g^−1^ FW, respectively. The drought-resistant accession VA14 had the maximum manganese (16.81 µg g^−1^ FW), copper (2.29 µg g^−1^ FW), zinc (14.63 µg g^−1^ FW), boron (7.44 µg g^−1^ FW), sodium (80.32 µg g^−1^ FW), and molybdenum (0.58 µg g^−1^ FW) contents. VA16 and VA12 had the next highest contents of these microelements, albeit accession VA16 had the maximum iron content (17.41 µg g^−1^ FW), followed by VA14 (16.77 µg g^−1^ FW). In contrast, accession VA3 had the lowest iron (10.26 µg g^−1^ FW), copper (0.98 µg g^−1^ FW), manganese (10.23 µg g^−1^ FW), sodium (62.55 µg g^−1^ FW), and (10.58 µg g^−1^ FW) zinc contents; and accession VA12 had the lowest molybdenum (0.26 µg g^−1^ FW) and boron (5.18 µg g^−1^ FW) contents. The amaranth leaves had greater iron and zinc contents compared with the cassava leaves and beach peas. The present investigation revealed ample iron, manganese, copper, zinc, sodium, molybdenum, and boron. Literature on several amaranth species [63] has shown ample manganese, copper, iron, zinc, molybdenum, boron, and sodium in several species of amaranths. They also mentioned that amaranth’s concentrations of manganese, copper, iron, and zinc were much higher than those in the spider flower, kale, spinach, and black nightshade.

The potassium and calcium found in drought-resistant accession VA14 were more plentiful than in green morph amaranth [60] and weedy amaranth [59]. The magnesium content recorded in the drought-resistant accession VA16 was greater than in green morph amaranth [60]. The phosphorus content of VA14 was higher than in weedy amaranth [59], whereas the sulfur contents of VA14 and VA16 were similar to that of our weedy amaranth [59]. The iron, zinc, and magnesium levels of VA14 and VA16 were much higher than in green morph amaranth [60] and *A. spinosus* (weedy amaranth) [59]. At the same time, the copper contents of the drought-resistant accessions VA14 and VA16 were higher than those in green morph amaranth [60]. The sodium and molybdenum contents of these drought-resistant accessions were higher than those in weedy amaranth [59]. In contrast, the boron levels of these drought-resistant accessions were similar to that found in our previous study of *A. spinosus* (weedy amaranth) [59]. Hence, VA14 and VA16 seem more mineral-enriched than the other amaranth accessions we tested.

### 3.3. Composition of Phytopigments

The pigment contents of four drought-resistant amaranths are shown in Figure 4. The drought-resistant amaranths had ample betalains, betaxanthins, and betacyanins, though the amounts varied significantly among the drought-resistant accessions (462.99 to 1007.09, 228.66 to 505.35, and 234.33 to 502.13 ng g^−1^ FW, respectively). The drought-resistant accessions VA14 and VA3 exhibited the maximum betacyanin contents (502.13 and 501.73 ng g^−1^ FW, respectively), followed by VA16 (484.55 ng g^−1^ FW). Conversely, the drought-resistant accession VA12 had the lowest betacyanin content (234.33 ng g^−1^ FW). Among the drought-resistant accessions, significant and pronounced variations were observed in betaxanthin content. The drought-resistant accession VA3 had the highest betalain and betaxanthin contents (1007.09 and 505.35 ng g^−1^ FW), and these values are statistically similar to those of drought-resistant accession VA14. In contrast, the drought-resistant accession VA12 showed the lowest betaxanthin and betalain contents (228.66, 462.99 ng g^−1^ FW). Chlorophyll *a* displayed significant and notable variations among accessions (132.45 to 518.16 µg g^−1^ FW). The drought-resistant accession VA14 had the maximum chlorophyll *a* content (518.16 µg g^−1^ FW), whereas the lowest chlorophyll *a* content was recorded in the drought-resistant accession VA12 (132.45 µg g^−1^ FW). Likewise, marked and significant differences in chlorophyll *b* content were observed in the drought-resistant amaranth accessions (72.55 to 264.13 µg g^−1^ FW). The drought-resistant accession VA16 had the maximum chlorophyll *b* content (264.13 µg g^−1^ FW), followed by the drought-resistant accession VA14 (254.58 µg g^−1^ FW). Conversely, the drought-resistant accession VA12 had the lowest chlorophyll *b* (72.55 µg g^−1^ FW). The drought-resistant accession of amaranth exhibited a significant and noticeable variation in total chlorophyll content (205.00 to 771.49 µg g^−1^ FW). The drought-resistant accessions VA14 and VA16 exhibited ample total chlorophyll contents, whereas the drought-resistant accession VA12 had the lowest total chlorophyll content (205.00 µg g^−1^ FW). Carotenoids showed significant and notable variations among accessions (66.58 to 138.79 mg 100 g^−1^ FW). The drought-resistant accession VA14 had the maximum carotenoid content (138.79 mg 100 g^−1^ FW), whereas the lowest carotenoid was recorded in the drought-resistant accession VA12 (66.58 mg 100 g^−1^ FW).

In this investigation, we reported ample total chlorophyll (771.59 µg g^−1^ FW), chlorophyll *a* (518.16 µg g^−1^ FW), chlorophyll *b* (264.13 µg g^−1^ FW), and carotenoid content (138.79 mg 100 g^−1^ FW) in the drought-resistant amaranths. Comparatively less chlorophyll content was observed in another study of *A. tricolor* by other authors [64]. In contrast, the same authors in different studies [64] observed corroborative results in red and green amaranth for betacyanins, betalains, chlorophylls, and betaxanthins. The drought-resistant accession VA14 and VA16 had ample betacyanins, betalains, chlorophylls, carotenoids, and betaxanthins content, indicating the presence of high antioxidant activity. The drought-resistant accessions VA14 and VA16 had ample betalains, chlorophylls, betacyanins, carotenoids, and betaxanthins, which have significant radical-quenching abilities. The presence of large amounts of betalains, betacyanins, carotenoids, chlorophylls, and betaxanthins in VA14 and VA16 may play a vital role in the quenching of ROS, and regular consumption of amaranth prevents many degenerative human diseases and counteract aging [22]. Contents of chlorophyll *A*, betalains, total chlorophyll, betacyanins, chlorophyll *b*, carotenoids, and betaxanthins obtained from drought-resistant accessions VA14, VA16, and VA3 were superior to those of red morph amaranth [58], green morph amaranth [60], stem amaranth [61], weedy amaranth [59], and *A. blitum* [62]. Hence, these selected drought-resistant accessions can use as high antioxidant-enriched accessions.

### 3.4. Bioactive Phytochemicals and Capacity to Quench Radicals

The drought-resistant amaranth varied markedly and significantly in polyphenol content, beta-carotene content, flavonoid content, ascorbic acid content, and antioxidant capacity (AC) (Figure 5). The beta-carotene in the drought-resistant accessions had pronounced variations (52.42 mg 100 g^−1^ FW in the drought-resistant accession VA3 to 81.25 mg 100 g^−1^ FW in the drought-resistant accession VA16). The ascorbic acid content of the drought-resistant accessions varied from 16.54 mg 100 g^−1^ FW in the drought-resistant accession VA3 to 158.54 mg 100 g^−1^ FW in the drought-resistant accession VA14. Polyphenols ranged from 66.54 (VA3) to 184.25 µg GAE g^−1^ FW (VA14). The drought-resistant accession VA14 had the maximum polyphenol content, followed by the drought-resistant accession VA16. The flavonoid content exhibited ample variation regarding the drought-resistant amaranth accessions (112.38 µg RE g^−1^ DW in the drought-resistant accession VA3 to 335.47 µg RE g^−1^ DW in the drought-resistant accession VA12). The AC (DPPH) of the drought-resistant amaranth ranged from 25.86 (VA12) to 32.34 µg TEAC g^−1^ DW (VA3).

Drought-resistant accession VA3 displayed the maximum AC (DPPH), followed by VA16 and VA14. In contrast, the drought-resistant accession VA12 had the lowest AC (DPPH). The AC (ABTS^+^) of the drought-resistant amaranths differed from 47.88 to 58.75 µg TEAC g^−1^ DW. The drought-resistant amaranth genotype VA3 had the maximum AC (ABTS^+^), followed by VA16 and VA14. Conversely, the drought-resistant accession VA12 had the minimum AC (ABTS^+^). These results are corroborative of the results of red and green amaranth. They obtained higher flavonoid and polyphenol contents, and AC in the red amaranth genotype compared with green amaranth. The drought-resistant accessions VA14 and VA16 showed high ascorbic acid, beta-carotene, and polyphenol levels. In contrast, the drought-resistant accessions of amaranth VA12 and VA14 had high flavonoids. The drought-resistant accessions of amaranth VA3, VA14, and VA16 had high AC in both DPPH and ABTS^+^ tests. Hence, these antioxidant phytochemicals of the drought-resistant accessions of amaranth VA3, VA14, and VA16 may be helpful for customers, and perform a key role in quenching ROS, thereby preventing aging and many destructive human diseases [22]. The drought-resistant accessions of amaranth VA3, VA14, and VA16 contained antioxidant phytochemicals such as phytopigments, ascorbic acid, beta-carotene, polyphenols, and flavonoids that have significant quenching abilities [1]. The beta-carotene contents obtained from these drought-resistant accessions VA14, VA16, and VA3 were greater than those of weedy amaranth [59], stem amaranth [61], and *A. blitum* [62]. The contents of ascorbic acid, polyphenols, and flavonoids; antioxidant capacities with DPPH; and antioxidant capacities with ABTS^+^ of VA14, VA16, and VA3 were greater than those of red morph amaranth [58], green morph amaranth [60], weedy amaranth [59], stem amaranth [61], and *A. blitum* [62]. Hence, drought-resistant accessions may have high levels of polyphenols and vitamin C. Beta-carotene, flavonoids, and antioxidants were enriched in these accessions compared with our previous amaranth accessions.

Ample phytopigments, including betalains, betacyanins, betaxanthins, chlorophylls, and carotenoids; bioactive compounds, such as ascorbic acid, beta-carotene, and phenols; quenching capacity; and flavonoids were observed in the drought-resistant amaranth accessions. The current results corroborate the levels of betacyanins, AC, betalains, flavonoids, betaxanthins, carotenoids, and polyphenols in other amaranth genotypes [64]. Betalains (1007.09 ng g^−1^), betacyanins (502.13 ng g^−1^), flavonoids (335.47 RE µg g^−1^ DW), betaxanthins (505.35 ng g^−1^), total chlorophyll (771.59 µg g^−1^ FW), beta-carotene (81.25 mg 100 g^−1^ FW), chlorophyll *b* (264.13 µg g^−1^ FW), carotenoids (138.79 mg 100 g^−1^ FW), chlorophyll *a* (518.16 µg g^−1^ FW), AC (ABTS^+^) (58.75 TEAC µg g^−1^ DW), ascorbic acid (158.54 mg 100 g^−1^ FW), polyphenols (184.25 GAE µg g^−1^ FW), and AC (DPPH) (32.34 TEAC µg g^−1^ DW) found in the current study corroborate the results in *A. tricolor* of literature [65], but the phenolics of the current study were much pronounced than in the *A. tricolor* of the previous study [65]. The drought-resistant accessions of amaranth VA3, VA14, and VA16 had high levels of ascorbic acid, betacyanins, carotenoid, betalains, betaxanthins, beta-carotene, chlorophylls, polyphenols, AC, and flavonoids. VA3, VA14, and VA16 could be used as high-yielding antioxidant-enriched varieties. These drought-resistant accessions contained adequate beta-carotene, polyphenols, ascorbic acid, flavonoids, phytopigments, and antioxidant potential, giving them prospects for consumption as nutraceuticals for the antioxidant-deficient community.

There were plentiful carbohydrates, protein, nutraceuticals, moisture, and dietary fiber; ample phytopigments, such as betacyanins, chlorophylls, betalains, carotenoids, and betaxanthins; adequate antioxidant phytochemicals, such as ascorbic acid, beta-carotene, polyphenols, and flavonoids; and antioxidant potential in the drought-resistant amaranth accessions. These results corroborate those of AC, flavonoids, betacyanins, carotenoids, betalains, polyphenols, and betaxanthins in red amaranth [65]. Environmental stress increases oxidative stress [66] which also increases the accumulation of phenolic compounds [9,12]. VA3, VA14, and VA16 had ample protein, carbohydrates, nutraceuticals, moisture, phytopigments, dietary fiber, flavonoids, and antioxidant potential. These drought-resistant accessions, VA3, VA14, and VA16, can be used as high-yielding antioxidant-enriched varieties.

### 3.5. Analysis of Correlation Coefficient

The correlations among antioxidant phytopigments and bioactive phytochemicals of the drought-resistant amaranths are shown in Table 1. The carotenoids, betalains, betaxanthins, total chlorophyll, betacyanins, chlorophyll *b*, and chlorophyll *a* displayed highly significant positive associations among each other and with AC (ABTS^+^ and (DPPH), ascorbic acid, beta-carotene, flavonoids, and polyphenols, and vice versa. Phytopigments of amaranth showed strong radical quenching ability, as all the phytopigments displayed significant correlations with AC (ABTS^+^ and DPPH). Beta-carotene and ascorbic acid showed significant correlations with all other components, along with AC (ABTS^+^ and DPPH).

Our results for ascorbic acid are corroborative of the results of Jimenez-Aguilar and Grusak [63] in various amaranth species. The significant positive associations of ascorbic acid and beta-carotene with AC (ABTS^+^ and DPPH) also suggest enormous antioxidant activity. Polyphenols and flavonoids were significantly associated with AC (ABTS^+^ and DPPH), indicating the strong antioxidant capacity of phenolics and flavonoids in vegetable amaranth. The TPC, TFC, and TAC (FRAP) of drought-induced amaranth corroborate the results of amaranth [5,67]. Similarly, the significant relationship of AC (ABTS^+^) with AC (DPPH) validated antioxidant activity measurements using different methods in vegetable amaranth.

## 4. Conclusions

Drought-resistant amaranth accessions had sufficient proximate, phytopigments, phytochemicals, nutraceuticals—including carbohydrates (8.38 g 100 g^−1^ FW)—moisture (88.16 g 100 g^−1^), protein (6.21 g 100 g^−1^), polyphenols (184.25 µg GAE g^−1^ FW), ascorbic acid (158.54 mg 100 g^−1^ FW), betaxanthins (505.35 ng g^−1^ FW), minerals—such as calcium (3.28 mg g^−1^ FW), phosphorus (1.81 mg g^−1^ FW), sulfur (1.34 mg g^−1^ FW), potassium (7.56 mg g^−1^ FW), magnesium (3.27 mg g^−1^ FW), iron (17.41 µg g^−1^ FW), copper (2.29 µg g^−1^ FW), zinc (14.63 µg g^−1^ FW), boron (7.44 µg g^−1^ FW), manganese (16.81 µg g^−1^ FW), molybdenum (0.58 µg g^−1^ FW), and sodium (80.32 µg g^−1^ FW)—dietary fiber (7.82 g 100 g^−1^ FW), beta-carotene (81.25 mg 100 g^−1^ FW), flavonoids (335.47 µg RE g^−1^ DW), betacyanins (502.13 ng g^−1^ FW), and chlorophylls—such as chlorophyll *a* (518.16 µg g^−1^ FW), chlorophyll *b* (264.13 µg g^−1^ FW), total chlorophyll (771.49 µg g^−1^ FW), and carotenoids (138.79 mg 100 g^−1^ FW)—AC (DPPH and ABTS^+^) (32.34 and 58.75 µg TEAC g^−1^ DW), and betalains (1007.09 ng g^−1^ FW). The drought-resistant amaranth accessions VA3, VA14, and VA16 had ample proximate, nutraceuticals, phytopigments, antioxidant phytochemicals, and antioxidant activity. The correlations revealed that all constituents of antioxidant potential of the drought-resistant amaranth accessions had strong radical quenching abilities. Hence, drought-resistant accessions VA3, VA14, and VA16 displayed admirable amounts of nutraceuticals, proximate, phytopigments, and antioxidant phytochemicals, along with radical quenching ability and ROS scavenging. These features offer enormous prospects for the promotion of health benefits and the feeding of communities in drought-prone semiarid and arid areas of the globe, especially those deficient in nutraceuticals, phytopigments, and antioxidants. VA3, VA14, and VA16 could be utilized as phytochemical-enriched cultivars in the drought-prone semiarid and arid areas of the globe.

## Figures and Tables

**Figure 1 antioxidants-11-00578-f001:**
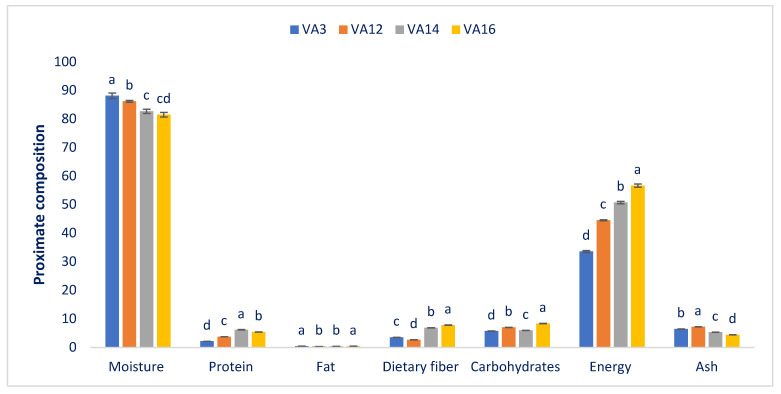
The moisture, protein, fat, dietary fiber, carbohydrate, energy, and ash contents of drought-resistant amaranth (g 100 g^−1^ FW). Dissimilar letters indicate statistical significance according to the Duncan Multiple Range Test (DMRT) (*p* < 0.01), (*n* = 3).

**Figure 2 antioxidants-11-00578-f002:**
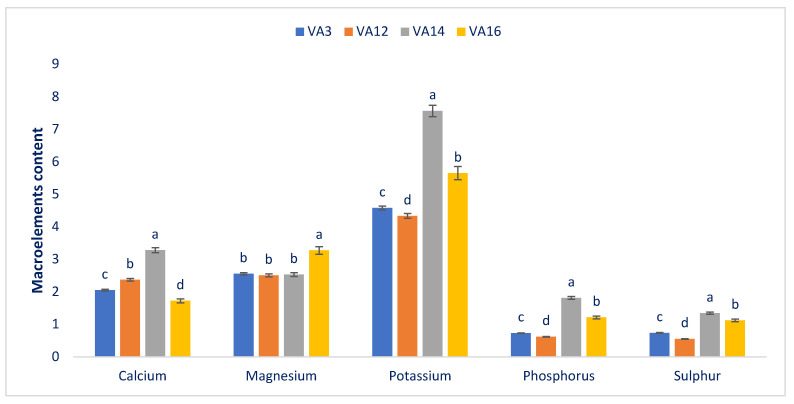
The microelement contents (mg g^−1^ FW) in four drought-resistant accessions of amaranth. Dissimilar letters indicate statistical significance according to (DMRT) (*p* < 0.01), (*n* = 3).

**Figure 3 antioxidants-11-00578-f003:**
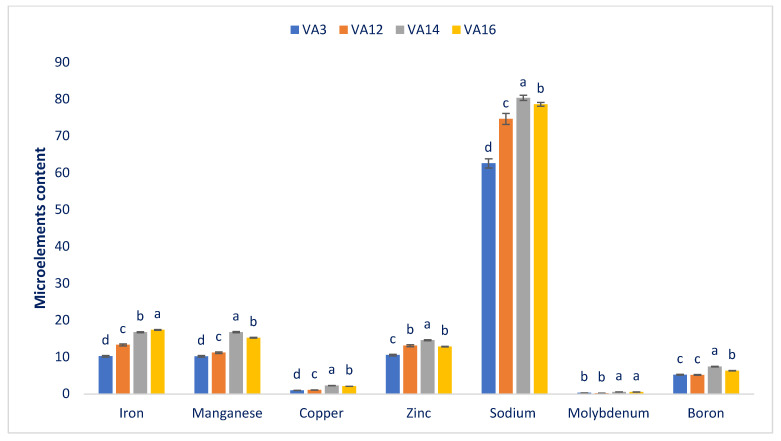
The microelement contents (µg g^−1^ FW) in the four drought-resistant accessions of amaranth. Dissimilar letters indicate statistical significance according to DMRT (*p* < 0.01), (*n* = 3).

**Figure 4 antioxidants-11-00578-f004:**
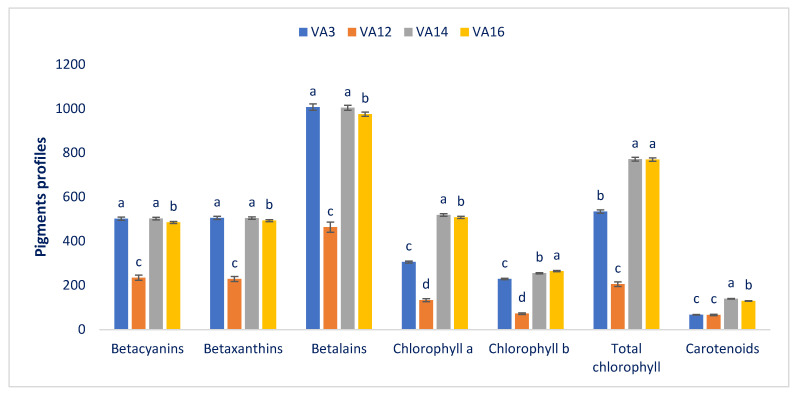
Bioactive phytopigment contents of drought-resistant amaranth (betacyanins (ng g^−1^ FW), chlorophyll *a* (µg g^−1^ FW), betaxanthins (ng g^−1^ FW), chlorophyll *b* (µg g^−1^ FW), betalains (ng g^−1^ FW), total chlorophyll (µg g^−1^ FW), total carotenoids (mg 100 g^−1^ FW)). Dissimilar letters indicate statistical significance according to DMRT (*p* < 0.01), (*n* = 3).

**Figure 5 antioxidants-11-00578-f005:**
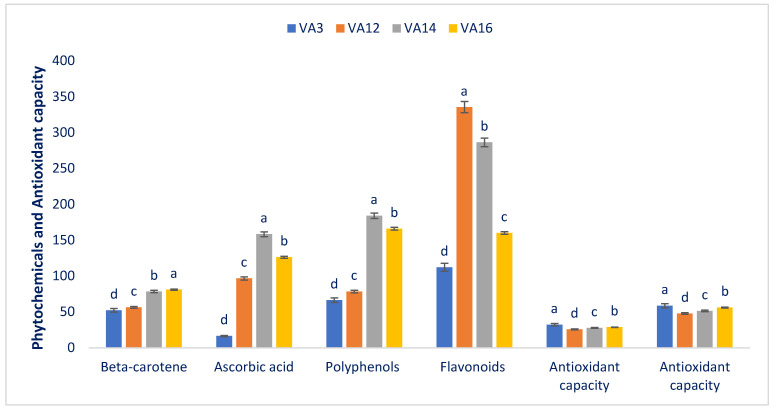
Contents of bioactive phytochemicals and free-radical scavenging capacities of drought-resistant amaranth (beta-carotene (mg 100 g^−1^ FW); ascorbic acid (mg 100 g^−1^ FW); polyphenols (µg GAE g^−1^ FW); flavonoids (µg RE g^−1^ DW); AC (DPPH), antioxidant capacity (DPPH) (µg TEAC g^−1^ DW); AC (ABTS^+^), antioxidant capacity (ABTS^+^) (µg TEAC g^−1^ DW)). Dissimilar letters indicate statistical significance according to DMRT (*p* < 0.01), (*n* = 3).

**Table 1 antioxidants-11-00578-t001:** The correlation coefficients for phytopigments, antioxidant capacity, and phytochemicals in drought-resistant vegetable amaranths.

	Bx	Bl	Chl *a*	Chl *b*	T. Chl	TC	BC	AA	TP	TF	AC (DPPH)	AC (ABTS^+^)
Bcn	0.95 **	0.88 **	0.94 **	0.96 **	0.92 **	0.88 **	0.85 **	0.93 **	0.92 **	0.91 **	0.98 **	0.89 **
Bx		0.86 **	0.98 **	0.98 **	0.95 **	0.87 **	0.87 **	0.96 **	0.91 **	0.85 **	0.81 **	0.93 **
Bl			0.97 **	0.89 **	0.96 **	0.84 **	0.94 **	0.87 **	0.85 **	0.93 **	0.96 **	0.98 **
Chl *a*				0.92 **	0.98 **	0.86 **	0.85 **	0.91 **	0.87 **	0.91 **	0.82 **	0.95 **
Chl *b*					0.93 **	0.86 **	0.82 **	0.87 **	0.88 **	0.83 **	0.93 **	0.92 **
T. Chl						0.88 **	0.78 **	0.85 **	0.94 **	0.94 **	0.89 **	0.94 **
TC							0.83 **	0.88**	0.92 **	0.91 **	0.94 **	0.97 **
BC								0.96**	0.95 **	0.89 **	0.87 **	0.85 **
AA									0.85 **	0.85 **	0.95 **	0.95 **
TP										0.86 **	0.85 **	0.97 **
TF											0.88 **	0.97 **
AC (DPPH)												0.98 **

Bcn = betacyanins (ng g^−1^), Bx = betaxanthins (ng g^−1^), Bl = betalains (ng g^−1^), Chl *a* = chlorophyll *a* (µg g^−1^), Chl *b* = chlorophyll *b* (µg g^−1^), T. chl = total chlorophyll (µg g^−1^), TC = total carotenoids (mg 100 g^−1^ FW), BC = beta-carotene (mg 100 g^−1^ FW), AA = ascorbic acid (mg 100 g^−1^ FW), TP = total polyphenols (GAE µg g^−1^ FW), TF = total flavonoids (RE µg g^−1^ DW), AC (DPPH) = antioxidant capacity (DPPH) (TEAC µg g^−1^ DW), AC (ABTS^+^) = antioxidant capacity (ABTS^+^) (TEAC µg g^−1^ DW), ** significant at 1% level, (*n* = 3).

## Data Availability

Data recorded in the current study are available in the tables and figures of the manuscript.

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
