# Peer review of "Bioactive Phytochemicals and Quenching Activity of Radicals in Selected Drought-Resistant Amaranthus tricolor Vegetable Amaranth"

_antioxidants, 2022, doi:10.3390/antiox11030578_

Round 1
Reviewer 1 Report
The authors report a phytochemical analysis of some secondary metabolites present in drought-resistant cultivars of Amaranthus tricolor. They determined the content of minerals, flavonoids and polyphenols and the antioxidant action of the plant extract.
Their manuscript therefore describes the components present in the extracts without correlating it to particular environmental conditions or scientific problems. Indeed, although the manuscript is well written and linear, unfortunately it is purely descriptive.
Furthermore, some fundamental passages are missing for a "phytochemical" manuscript: analytical procedures that evaluate the semi-quantitative content of the phytoextracts. The techniques used for the determination of secondary metabolites are still very archaic.
Another aspect that should be better discussed is the extraction procedure. The authors did not report the protocol applied for the extraction. Did they use the roots, leaves or other parts of the plant?
I suggest that the authors include this manuscript in the references:
Kumar A et al. Plant behaviour: an evolutionary response to the environment? Plant Biol (Stuttg). 2020 Nov;22(6):961-970. doi: 10.1111/plb.13149.
Author Response
Reviewer 1
Comment: The authors report a phytochemical analysis of some secondary metabolites present in drought-resistant cultivars of Amaranthus tricolor. They determined the content of minerals, flavonoids and polyphenols and the antioxidant action of the plant extract.
Author response: We would like to appreciate and thank honorable Reviewer 1 for giving valuable time and critically reviewing our MS for its substantial improvement and also for giving the positive decision to publish.
Comment: Their manuscript therefore describes the components present in the extracts without correlating it to particular environmental conditions or scientific problems. Indeed, although the manuscript is well written and linear, unfortunately it is purely descriptive.
Author response: I again would like to appreciate and thank honorable Reviewer 1 for the positive comments on manuscript writing. We try our best to concise the manuscript according to suggestions of you and reviewer 2 (please see the red color sentences in the revised MS).
Comment: Furthermore, some fundamental passages are missing for a "phytochemical" manuscript: analytical procedures that evaluate the semi-quantitative content of the phytoextracts. The techniques used for the determination of secondary metabolites are still very archaic.
Author response: We agree with you that the techniques we followed are archaic. However, till today, these are popular and the most used techniques for a researcher for the determination of these phytochemicals. There are several thousand high-ranked and high-impact factor WOS articles in literature where researchers followed these techniques. For your kind consideration, our research group published many articles including food chemistry (IF-7.514) utilizing these techniques. I am giving you very few references, doi.10.1016/j.foodchem.2018.01.097; doi.10.1186/s12870-018-1484-1; doi.10.3389/fpls.2020.559876.
Comment: Another aspect that should be better discussed is the extraction procedure. The authors did not report the protocol applied for the extraction. Did they use the roots, leaves, or other parts of the plant?
Author response: We followed protocols from previously published articles for the extraction and cited references of each protocol from where we adopted. For this reason, we briefly describe the determination procedures. We used the leaf sample for extraction which was written at the time of the first submission. Now, I have marked them in red color.
Comment: I suggest that the authors include this manuscript in the references:
Kumar A et al. Plant behaviour: an evolutionary response to the environment? Plant Biol (Stuttg). 2020 Nov;22(6):961-970. doi: 10.1111/plb.13149.
Author response: We have added the suggested reference (please see ref 66 marked by red color).
Reviewer 2 Report
The review of submitted revised manuscript entitled "Bioactive phytochemicals and quenching activity of radicals in selected drought-resistant Amaranthus tricolor amaranth"
The authors have attached the text signed ...as its revised version for its reassessment.
- Promising title, but Abstract lacks any data obtained. Abstract can not a story about the research. Please, refine and shorten this part of the manuscript.
- The keywords seem to be well chosen. The Introduction is well written, with the only shortcoming: the authors did not formulate a research hypothesis in the last paragraph (which they should have done).
- Materials and Methods: In principle , I have no objections here. Two minor issues are A) The form of recording Hitachi, Japan - what type of device and the need to unify the spelling (lines 152, 160, 169, 186, 197), B) One way analysis of variance - put One-way on the beginning of line 223.
- Results and Discussion: This is an essential part of the manuscript, but unfortunately I am unable to accept its current form. I'll try to explain why. First, I have never encountered such an inappropriate and completely schematic presentation of results throughout the text (see: lines 257-272, 285-301, 312-336). Second, this is not properly written discussion. From what I understand the authors mainly were discussed themselves. Please, address this objection! Third, Figures captions are too poor. The caption of figure must definitely be precise and unambiguous. (Fig. 1: The composition of exactly what? Fig. 2 (perhaps it would be better:) The content of macroelements in four drough-resistant lines/accesions of amaranths..... The same comments apply to the captions of all figures.
- References are not adequate in many cases for this manuscript. At least about 10 items should be removed.
- Conclusions: This element of manuscript needs to be rewritten. Please, Do read carefully the attached sentences. They have nothing to do with the conclusion of the experiment performed here.
It occurs to me that the manuscript should be withdrawn. Please, think carefully how to correct the indicated imperfections.
Author Response
Reviewer 2
Comment: The review of submitted revised manuscript entitled "Bioactive phytochemicals and quenching activity of radicals in selected drought-resistant Amaranthus tricolor amaranth"
The authors have attached the text signed ...as its revised version for its reassessment.
Author response: We would like to appreciate and thank honorable Reviewer 2 for giving valuable time and critically reviewing our MS for its substantial improvement and also for giving the positive decision to publish. The manuscript is not peer-reviewed earlier. The manuscript returned to rephrase a few sentences for minimizing plagiarism. For this reason, we mention a revised version.
Comment:
- Promising title, but Abstract lacks any data obtained. Abstract can not a story about the research. Please, refine and shorten this part of the manuscript.
Author response: We appreciate your comments. We revised the abstract adding some results without any values (please see red color sentences in the revised MS) and concise to shorten its content.
Comment:
- The keywords seem to be well chosen. The Introduction is well written, with the only shortcoming: the authors did not formulate a research hypothesis in the last paragraph (which they should have done).
Author response: We appreciate your comments. The hypothesis of the research has been added in the last paragraph before the objectives of the study (please see the red color sentence in the revised MS).
Comment:
- Materials and Methods: In principle, I have no objections here. Two minor issues are A) The form of recording Hitachi, Japan - what type of device and the need to unify the spelling (lines 152, 160, 169, 186, 197), B) One way analysis of variance - put One-way on the beginning of line 223.
Author response: We appreciate your comments. A) we have mentioned the type of device and unified the spelling (please see lines 139, 151, 166, 186, 198, 208, 214 in the revised MS). B) we added the word “One-way” at the beginning of the sentence (please see line 224 in the revised MS).
Comment:
- Results and Discussion: This is an essential part of the manuscript, but unfortunately I am unable to accept its current form. I'll try to explain why. First, I have never encountered such an inappropriate and completely schematic presentation of results throughout the text (see: lines 257-272, 285-301, 312-336). Second, this is not properly written discussion. From what I understand the authors mainly were discussed themselves. Please, address this objection! Third, Figures captions are too poor. The caption of figure must definitely be precise and unambiguous. (Fig. 1: The composition of exactly what? Fig. 2 (perhaps it would be better:) The content of macroelements in four drough-resistant lines/accesions of amaranths..... The same comments apply to the captions of all figures.
Author response: According to your suggestions we have rewritten the sentences in previous lines 257-272, 285-301, 312-336 (please see lines 260-270, 284-292, 304-314 in the revised MS). There is scarce literature regarding this type of work in amaranth except for a few works of our research group and others that we already cited in different sections. The figure captions have been changed following your suggestions (please see red color sentences in the revised MS).
Comment:
- References are not adequate in many cases for this manuscript. At least about 10 items should be removed.
Author response: We have removed 10 references from the manuscript. The reference number is now reduced from 75 to 65, however, according to suggestions of reviewer 1, we added ref number 66.
Comment:
- Conclusions: This element of manuscript needs to be rewritten. Please, Do read carefully the attached sentences. They have nothing to do with the conclusion of the experiment performed here.
Author response: According to your suggestions we have rewritten the conclusion by adding conclusive remarks/recommendations of the study (please see red color sentence in the revised MS).
Round 2
Reviewer 1 Report
The manuscript has improved considerably; the authors have finalized their paper following the indications suggested by the reviewers
I am very satisfied with the corrections and additions made by the authors.
In my opinion, the manuscript can now be published.
Author Response
Reviewer 1 (round 2)
Comments:
The manuscript has improved considerably; the authors have finalized their paper following the indications suggested by the reviewers
I am very satisfied with the corrections and additions made by the authors.
In my opinion, the manuscript can now be published.
Author response:
We would like to appreciate and thank honorable Reviewer 1 for giving valuable time and critically reviewing our MS for its substantial improvement and also for giving the positive decision to publish.
Reviewer 2 Report
I am pleased to say that in my opinion the authors have made an effort to improve their manuscript. In principle, it would be ready for further editorial work, but I leave the option of minor revision to the corresponding author so that he could remove one of the (two occurring side by side) words "content" from line 306. In the conclusions subsection one could present some obtained numerical values, those particularly noteworthy from the author's point of view.
Author Response
Reviewer 2 (round 2)
Comments
I am pleased to say that in my opinion the authors have made an effort to improve their manuscript. In principle, it would be ready for further editorial work, but I leave the option of minor revision to the corresponding author so that he could remove one of the (two occurring side by side) words "content" from line 306. In the conclusions subsection one could present some obtained numerical values, those particularly noteworthy from the author's point of view.
Author response
We would like to appreciate and thank honorable Reviewer 2 for giving valuable time and critically reviewing our MS for its substantial improvement and also for giving the positive decision to publish.
The one of the duplicate word “contents” have been removed from the 306. For your kind consideration, if we add numerical values against each trait in the conclusion section, it will be very large. Hence, we didn’t include numerical values.